# Prevalence of CAH-X Syndrome in Italian Patients with Congenital Adrenal Hyperplasia (CAH) Due to 21-Hydroxylase Deficiency

**DOI:** 10.3390/jcm11133818

**Published:** 2022-07-01

**Authors:** Rosa Maria Paragliola, Alessia Perrucci, Laura Foca, Andrea Urbani, Paola Concolino

**Affiliations:** 1Unit of Endocrinology, Agostino Gemelli Foundation University Hospital IRCCS, 00168 Rome, Italy; rosamaria.paragliola@unicatt.it; 2Department of Translational Medicine and Surgery, Catholic University of Sacred Heart, 00168 Rome, Italy; 3UniCamillus-Saint Camillus International University of Health Sciences, 00131 Rome, Italy; 4Clinical Chemistry, Biochemistry and Molecular Biology Operations (UOC), Agostino Gemelli Foundation University Hospital IRCCS, 00168 Rome, Italy; alesperrucci@gmail.com (A.P.); laura.foca@guest.policlinicogemelli.it (L.F.); andrea.urbani@policlinicogemelli.it (A.U.); 5Department of Basic Biotechnological Sciences, Intensivological and Perioperative Clinics, Catholic University of Sacred Heart, 00168 Rome, Italy

**Keywords:** congenital adrenal hyperplasia, Ehlers–Danlos syndrome, CAH-X syndrome

## Abstract

21-hydroxylase deficiency (21OHD), the most common form of congenital adrenal hyperplasia (CAH), is associated with pathogenic variants in *CYP21A2* gene. The clinical form of the disease ranges from classic or severe to non-classic (NC) or mild late onset. The *CYP21A2* gene is located on the long arm of chromosome 6, within the RCCX region, one of the most complex loci in the human genome. The 3′untranslated sequence of *CYP21A2* exon 10 overlap the last exon of *TNXB* gene (these genes lie on the opposite strands of DNA and have the opposite transcriptional direction) that encodes an extracellular matrix glycoprotein tenascin-X (TNX). A recombination event between *TNXB* and its pseudogene *TNXA* causes a 30 kb deletion producing a chimeric *TNXA/TNXB* gene (CAH-X chimera) where both *CYP21A2* and *TNXB* genes are impaired. This genetic condition characterizes a subset of patients with 21OHD who display the hypermobility phenotype of Ehlers–Danlos syndrome (hEDS) (CAH-X Syndrome). The aim of this study was to assess the prevalence of CAH-X syndrome in an Italian cohort of patients with 21OHD. At this purpose, 196 probands were recruited. Multiplex ligation-dependent probe amplification (MLPA) and Sanger sequencing were used to identify the CAH-X genotype. Twenty-one individuals showed the heterozygous continuous deletion involving the *CYP21A2* and part of the *TNXB* gene. EDS-related clinical manifestations were identified in most patients carrying the CAH-X chimera. A CAH-X prevalence of 10.7% was estimated in our population.

## 1. Introduction

The term CAH-X was coined to describe a subset of patients with congenital adrenal hyperplasia (CAH) who display the hypermobility phenotype of Ehlers–Danlos syndrome (hEDS) due to the monoallelic presence of a *CYP21A2* deletion extending into the *TNXB* gene [1,2]. Both *CYP21A2* and *TNXB* gene are located in RCCX region on chromosome 6 [3]. *CYP21A2* encodes the steroid 21-hydroxylase enzyme that drives aldosterone and cortisol biosynthesis. Deleterious variants in this gene induce 21-hydroxylase deficiency (21OHD), an autosomal recessive disease representing the most common cause of CAH [4,5]. The clinical form of the disease ranges from classic or severe to non-classic (NC) or mild late onset [6,7]. The classic CAH, showing a prevalence of 1:16,000 live births in the Caucasian population, is in turn classified into two forms on the basis of disease severity: the salt-wasting (SW), associated with *CYP21A2* variants removing enzyme activity, with deficiency of cortisol and aldosterone, and the simple virilizing (SV), associated with *CYP21A2* variants retaining <5% of enzyme activity, with ability to synthetize aldosterone. On the other hand, the NC (mild) form, with a prevalence in the range of 1:1000–1:500 live births, is caused by *CYP21A2* variants that retain 20–50% of enzyme activity. This form is often asymptomatic and the diagnosis is made during biochemical evaluations performed for hirsutism and menstrual irregularities [8,9].

The EDS has an estimated prevalence of about 1:5000 and represents one of the several heritable connective tissue disorders, characterized by a variable degree of skin hyperextensibility, joint hypermobility (JHM), and tissue fragility [10]. The last classification of the International EDS Consortium recognizes 13 subtypes with 19 different causal genes involved in collagen and extracellular matrix synthesis and maintenance [11]. The hEDS is the most common type of EDS and involves generalized joint hypermobility (GJH), frequent joint dislocations, arthralgias, hernias and midline defects including cardiac structural abnormalities [11]. It was reported that heterozygous *TNXB* pathogenic variants are related to hEDS phenotype [11,12,13]. In fact, the *TNXB* gene, spanning 68.2 kb and containing 44 exons, encodes the tenascin-X protein (TNX), an extracellular matrix glycoproteins which is highly expressed in connective tissue [14].

In the RCCX region, a recombination event between *TNXB* and its pseudogene *TNXA* causes a 30 kb deletion (involving the whole sequence of the *CYP21A2* gene and extending into the *TNXB* gene) producing a chimeric *TNXA/TNXB* gene (CAH-X chimera) where both *CYP21A2* and *TNXB* genes are impaired [1,3]. Different junction sites distinguish three *TNXA/TNXB* chimeras (CH-1, CH-2 and CH-3) [2,3]. CAH-X CH-1 is characterized by the presence, within exon 35, of a *TNXA* derived 120 bp deletion causing the non-functionality of *TNXB* gene by means of a haploinsufficiency mechanism [1,14,15]. In contrast, CH-2 and CH-3 chimeras, carrying missense pathogenic variants, are associated with a dominant-negative effect [15,16]. While a single missense *TNXB* variant (NM_019105.8):c.12174C>G; p.Cys4058Trp, exon 40) characterizes the CH-2 chimera, the CH-3 carries a cluster of three missense variants: the (NM_019105.8):c.12218G>A (p.Arg4073His) in exon 41, the (NM_019105.8):c.12514G>A (p.Asp4172Asn) and the (NM_019105.8):c.12524G>A (p.Ser4175Asn) in exon 43 [15,16]. All three CAH-X chimeras, in either monoallelic or biallelic form, cause hEDS and the clinical manifestations of disease appear to be more severe in patients with CAH [2].

The prevalence of CAH-X syndrome was estimated to be around 14–15% in three large cohort of patients with 21OHD [17,18,19]. Some authors proposed including CAH-X chimeras screening in routine genetic testing of 21OHD, as this could be beneficial to ensure a very early diagnosis to young children and to allow early detection of cardiac defects [20]. In response to these considerations, other authors emphasized the need of performing further studies for a comprehensive understanding of the CAH-X molecular background and to fully define the clinical picture of the syndrome [21,22]. In fact, population studies are still scarce, and data regarding the Italian population are currently absent.

Based on these observations, we aimed to investigate the prevalence of CAH-X syndrome in an Italian cohort of patients with 21OHD, performing the molecular characterization of *TNXA/TNXB* gene and the correlation between genotype and EDS-associated clinical symptoms.

## 2. Methods

### 2.1. Patients

We considered 196 unrelated patients with 21OHD (age ≥ 18), referred to our laboratory (Diagnostics Unit of Molecular Biology and Genomics, Fondazione Policlinico Universitario “A. Gemelli” IRCCS of Rome) between 2018 and 2021 in order to confirm the clinical diagnosis by the genetic testing. The signing of informed consent was required for participation in the study. The project was approved by the ethics committee of Fondazione Policlinico Universitario “A. Gemelli” IRCCS of Rome (ID:4632) and all procedures were carried out in accordance with the 1964 Helsinki declaration and its ethical standards.

### 2.2. Molecular Analysis of CYP21A2 Gene

Genomic DNA was isolated using High Pure PCR Template Preparation Kits (Roche Diagnostics, Indianapolis, IN, USA), quantified by a spectrophotometer at 260 nm and stored at −20 °C. *CYP21A2* analysis was performed by using polymerase chain reaction (PCR) combined with Sanger sequencing and the multiplex ligation-dependent probe amplification (MLPA) assay. Briefly, an 8.5 kb PCR product, covering part of the RCCX region of the chromosome 6, was amplified (using CYP779f 5′-ccagaaagctgactctggatg-3′ and Tena32F 5′-ctgtgcctggctatagcaagc-3′ primers) and directly sequenced using the BigDye Terminator Cycle Sequencing Kit, Version 3.1 (Applied Biosystems, Waltham, MA, USA) and an ABI 3100 Avant Genetic Analyser (Applied Biosystems) according to the manufacturer’s instructions (internal sequencing primers are available on request). Sequencing electropherograms were analyzed against the reference sequence NM_000500.9 using the SeqScape Version 3.0 software package (Applied Biosystems).

MLPA was employed to establish the exact copy number of the *CYP21A2* gene (SALSA MLPA Probemix P050 CAH by MRC Holland, Amsterdam, The Netherlands). Three healthy individuals were included in the analysis as controls. Results were analyzed by Coffalyser.NET Software (MRC Holland, Amsterdam, The Netherlands).

### 2.3. Detection of TNXA/TNXB Gene

The continuous deletion involving the whole *CYP21A2* gene and the 35th exon of *TNXB* (CH-1 chimera) was detected using MLPA method (SALSA MLPA Probemix P050 CAH by MRC Holland, Amsterdam, The Netherlands) according to the manufacturer’s instruction. In fact, MLPA kit contains two probes for the wild-type sequence of *TNXB* exon 35. When the *TNXA* derived 120 bp deletion is present, these probes fail to bind their target showing a final ratio (FR) <0.80 (normal range 0.80 < FR > 1.20) in heterozygous patients.

In contrast, to assess the presence of the other *TNXA/TNXB* chimeras, the 8.5 kb PCR was directly sequenced using specific primers for exons 40–44 of *TNXB* gene (sequences available on request). In fact, when the “30 kb deletion” involves the whole sequence of *CYP21A2* gene and part of the *TNXB* gene, the 8.5 kb PCR fragment contains the *CYP21A1P* pseudogene and the *TNXA/TNXB* chimeric gene. Sequencing electropherograms were analyzed against the *TNXB* reference sequence NM_001365276.2 using the SeqScape Version 3.0 software package (Applied Biosystems).

### 2.4. Assessment of EDS-Related Clinical Characteristics

In order to assess clinical manifestations of the hEDS, the same examiner performed physical examination and review of medical and family histories. The Beighton score, a set of maneuvers in a nine-point scoring system used as the standard method of assessment for generalized joint hypermobility (GJH), was employed to evaluate the mobility of the main joints, and GJH was defined as a score ≥ 5 [11,23,24,25]. Other joint manifestations (such as recurrent dislocation/subluxations and acute and chronic pain) and skin characteristics (extensibility, fragility, thinness, scar formation, oppressive papules and striae) were also valuated. Information regarding gastrointestinal disorders (gastroesophageal reflux, heartburn, bloating, recurrent abdominal pain, constipation, diarrhea and irritable bowel syndrome), morphological features (rectal prolapse, diaphragmatic hernias, abdominal hernias, ptosis of internal organs and intestinal intussusceptions), structural cardiac abnormalities and other hEDS related characteristics were acquired examining the medical history of the patient.

### 2.5. Statistical Analysis

The qualitative variables were described by absolute and percentage frequencies, while the quantitative variables were summarized with the most appropriate index in relation to their distribution, which was evaluated through the Shapiro–Wilk test. In particular, where normally distributed, they have been described by means of the mean and standard deviation (SD), otherwise by means of the median and interquartile range (IQR). In order to assess the differences in the occurrence of EDS-related clinical manifestations in the patients carrying different genotypes, the chi-square test by Fisher’s correction was used. STATA version 16 software package (STATA Corp) was employed for all statistical analysis, and the differences was considered statistically significant when P did not exceed 0.05.

## 3. Results

### 3.1. Characteristics of Patients

Among the 196 adult patients with 21OHD, 127 were female, with a median age of 23 (aged 18–53) years, including 62 cases of salt-wasting (SW) type, 21 cases of simple virilizing (SV) type, and 44 cases of NC type. The 69 males, with median age of 27 (aged 19–57) years, included 37 cases of SW type, 21 cases of SV type and 11 cases of NC type.

### 3.2. Molecular Analysis

In our cohort, the 30 kb deletion was identified in 74 patients (37.7%, 74/196). Among them, 21 (28.4%, 21/74) individuals carried the heterozygous continuous deletion involving the *CYP21A2* and part of the *TNXB* gene. In particular, MLPA assay detected the *TNXB* exon 35 deletion in 13 patients, which resulted in carriers of the CH-1 chimera. Seven individuals showed the c.1274C>G (p.Cys4058Trp) variant in the exon 40 of *TNXB* gene (CH-2 chimera), and only one had the cluster variant of exons 41 (c.12218G>A, p.Arg4073His) and 43 (c.12514G>A, p.Asp4172Asn and c.12524G>A, p.Ser4175Asn) representing the CH-3 chimera. No biallelic CAH-X forms were detected in our cohort. Based on these results, the frequencies of each of the three genotypes, CAH-X CH-1, CH-2 and CH3, were 6.6% (13/196), 3.6% (7/196), and 0.5% (1/196), respectively.

Detailed information about the genotype of the 21 patients with CAH-X are shown in Table 1.

### 3.3. Clinical Findings in Patients with CAH-X

EDS-related clinical manifestations were evaluated in all 21 patients with CAH-X and in some of their relatives carrying the continuous deletion involving the *CYP21A2* and part of the *TNXB* gene (Table 2, Table 3 and Table 4). Table 2 shows clinical findings in 13 unrelated probands with monoallelic CH1 CAH-X and 3 relatives that were available for molecular and clinical evaluation. All probands exhibited one or more characteristics of the hEDS phenotype. GJH and skin laxity was observed in 7/13 (53.8%) and 4/13 (30.8%) patients, respectively (Table 2). Cardiac defects, consisting of atrial septum and trivial mitral insufficiency, were found in 2/13 (15,4%) probands with monoallelic CH1 CAH-X (Table 2). Other clinical hEDS features were observed in 9/13 (69.2%) patients (Table 2). Three parents were available for clinical evaluation. They were carriers of CH1 chimera but did not have 21-hydroxylase deficiency (Table 2). One of them (BM-067 Mother) presented GJH (Beigthon score: 7) with no other features of the hEDS (Table 2). Proband BM-044’s mother was a woman with psychiatric problems and reported symptoms such as chronic fatigue, anxiety, depression and easy bruising. Finally, the proband BM-051′s father did not have any symptoms related to hEDS (Table 2).

Table 3 shows clinical findings in seven unrelated probands with monoallelic CH2 CAH-X and two relatives that were available for molecular and clinical evaluation. GJH was detected in 3/7 (42.8%) probands, while only one patient (BM-072) (1/7; 14.3%) presented thin skin with laxity (Table 3). No heart defect was found in patients who underwent echocardiography (Table 3). In addition, 71.4% (5/7) of the probands showed other clinical hEDS features (Table 3). The two examined parents did not show relevant features related to hEDS, and only the proband BM-005’s mother presented hallux valgus and gastroesophageal reflux as minor symptoms (Table 3).

Regarding GJH and skin involvement, no statistically significant difference was observed between CH1 and CH2 patients (Fisher exact test *p* = 1.00 for GJH and *p* = 0.6065 for skin laxity).

In our cohort, CH3 chimera was identified in only one patient (BM-037) with SV 21OHD (Table 4). The young boy presented with a significant hEDS phenotype, with a Beighton score of 9/9 indicating severe GJH. No skin defects were identified, while the echocardiography identified mitral regurgitations. In addition, the proband reported two episodes of ankle dislocation (Table 4). The mother and two sisters, all carriers of the CH3 chimera, were available for clinical evaluation: the older sister presented with a Beighton score of 7/9 while the mother and the young sister showed no signs of GJH. The mother had abdominal striae and piezogenic papules on both feet (Table 4).

## 4. Discussion

This is the first time that CAH-X molecular analysis is described in a cohort of Italian patients with 21OHD. We considered 196 unrelated patients previously screened by Sanger sequencing and MLPA assay in order to confirm the clinical diagnosis of 21OHD. Regarding the data obtained, we would like to first discuss the number of individuals carrying the “30 Kb deletion”, assuming that, in previous population studies, a different meaning of the “30-kb deletion” term was used [1,17,18,19]. In particular, while Merke et al. used the term “30-kb deletion” to refer to all chimeras (*CYP21A1P/CYP21A2* and *TNXA/TNXB*) identified in the RCCX region [17], Marino et al. and Gao et al. used this term to describe chimeras with junction sites downstream *CYP21A2* exon 7 and chimeras with junction sites downstream of *CYP21A2* exon 3, respectively [18,19]. In our cohort, 37.7% (74/196) of subjects resulted heterozygous for the “30-kb deletion” on the chromosome 6. Because we also considered the term “30-kb deletion” as inclusive of all RCCX chimeras, we compared our data with the result obtained in the last study of the NIH research group where 72 patients with the “30-kb deletion” were identified in a court of 135 individuals with 21OHD (72/135; 53%) [17]. This percentage of 53% is significantly different if compared to frequency detected in our population (37.7%) (*p* = 0.0068). In addition, although not statistically significant (*p* = 0.0645), a trend toward a higher prevalence of “30-kb deletion” in the NIH population continued when comparing our frequency (37.7%) with that obtained in a previous study (47.4%) by the same research group [1].

Looking at the number of patients carrying a chimeric *TNXA/TNXB* gene, we identified 21 individuals with the heterozygous continuous deletion among the 74 patients with the “30-kb deletion” (28.4%, 21/74). This result was in agreement with the data obtained in the last study by Merke et al. reporting 29.2% of patients with *TNXA/TNXB* gene among individuals with “30-kb deletion” (21/72) [17]. The frequencies obtained in the Argentine and Chinese populations were significantly higher, 73% and 62.8%, respectively [18,19].

In the present study, 10.7% (21/196) of patients with 21OHD were confirmed to have a continuous deletion involving *CYP21A2* gene and part of *TNXB* (CAH-X). This prevalence was less than that previously reported in the NIH cohort (15.6%) or in the Argentine and Chinese populations (14.2% and 13.9%, respectively) [17,18,19].

The prevalence of CAH-X CH-1, CH-2 and CH3, was 6.6% (13/196), 3.6% (7/196), and 0.5% (1/196), respectively. In agreement with other studies [17,18,19], also in our cohort, CH1 was the most recurrently detected chimera and CH3 the least frequent. In addition, no biallelic form of CAH-X were identified in our population.

Previous studies reported that most patients with CAH-X syndrome had mild to severe connective tissue symptoms, including GJH, recurrent dislocation of joints, hyperextensible skin and lesions of other organs [1,17,18,19]. Similarly, our study identified EDS-related clinical manifestations in most individuals carrying the *TNXA/TNXB* gene (Table 2, Table 3 and Table 4). However, in our cohort, the degree of articular hypermobility and skin manifestations of patients with CAH-X CH2 was not significantly different from that of patients carrying the CH1 chimera. Cardiac defects were present in two patients with monoallelic CH1 CAH-X and in the proband carrying the CH3 chimera. No heart defects were identified in patients with CH2 CAH-X (Table 2, Table 3 and Table 4). However, in our cohort, the prevalence of cardiac defects could be underestimated and asymptomatic cardiac diseases would be missed. In fact, this aspect was investigated by retrospectively analyzing the clinical history of the patient and the results of instrumental tests.

Few data regarding the phenotype associated with CAH-X CH3 chimera are available [9,11,18,19,26,27,28]. Marino et al. found the CH3 chimera in one monoallelic patient who was not available for clinical evaluation [18]. Recently, Gao et al. detected CH3 chimera in 11 patients; however, only one of them was available for clinical evaluation showing joint hypermobility and poor wound healing [19]. In our cohort, we identified CH3 chimera in only one patient with severe GJH and a cardiac defect due to mitral regurgitation.

A normal or very mild phenotype has been detected in the relatives of CAH-X patients who were carriers of one CAH-X allele, but not affected by CAH. Two of these were observed to have joint hypermobility, while others showed skin or gastrointestinal symptoms or were asymptomatic (Table 2, Table 3 and Table 4).

The association between hEDS and CAH in the CAH-X syndrome represents a clinical challenge. Connective tissue dysplasia should be evaluated in all CAH patients, especially in those harboring a deletion in the CYP21A2 gene. However, the possible coexistence of hEDS could be difficult to evaluate, considering the variability of clinical presentation and the need of an objective assessment tool to measure the GJH. Furthermore, a concomitant therapy with synthetic glucocorticoid, which can be used in CAH, should have an impact on some hEDS-associated symptoms, for example, masking the chronic pain. Therefore, possible consideration could be given about the opportunity to offer the genetic test for the CAH-X in patients affected by CAH, especially to guarantee a prompt diagnosis of the possible cardiac abnormalities associated with the hEDS phenotype [20]. This evaluation could be helpful in the clinical management, considering that patients affected by CAH present with increased prevalence of risk factors for cardiovascular diseases, including obesity, hypertension and insulin resistance [29]. On the other hand, it is questionable if patients affected by hEDS should be screened for late onset CAH, which represents an asymptomatic form of CAH. In fact, although the role of CAH-related hormonal pattern as well as of chronic glucocorticoid therapy in connective tissue dysplasia is not fully understood, the CAH-X patients are consistently more severely affected than patients with homozygous or heterozygous TNX-deficient-type EDS without CAH [28].

## 5. Conclusions

In this report, we assessed the prevalence of the chimeric *TNXA/TNXB* gene for the first time in an Italian cohort of patients with 21OHD. We found that about 11% of patients had the continuous deletion involving *CYP21A2* gene and part of *TNXB*, and most of them showed EDS-related clinical symptoms. These data provide a further contribution to improve our knowledge on *TNXB*-related disorders and to investigate the role of hormonal milieu in connective-tissue pathophysiology. In fact, *TNXB* and CAH-X studies are necessary to define detailed guidelines for the surveillance of clinical manifestations and other long-term complications. In this field, the utility to offer the genetic test to establish a prompt diagnosis of CAH-X should be carefully evaluated.

## Figures and Tables

**Table 1 jcm-11-03818-t001:** Genotype and clinical 21OHD phenotype in 21 patients with CAH-X.

Patient (Sex)	Age (Years)	*CYP21A2* Genotype (allele1/allele2)	21OHD Phenotype	*TNXA/TNXB* Chimera
BM-002 (F)	24	p.(Arg357Trp)/del	SW	CH1
BM-014 (F)	41	p.(Gln319Ter)/del	SW	CH1
BM-023 (M)	42	p.(Val282Leu)+ c.293-13A/C>G/del	SW	CH1
BM-044 (F)	18	c.293-13A/C>G/del	SW	CH1
BM-067 (M)	19	c.293-13A/C>G+ p.(Gln319Ter)/del	SW	CH1
BM-072 (F)	21	c.293-13A/C>G/del	SW	CH2
BM-077 (M)	37	p.(Pro31Leu)+ p.(Arg357Trp)/del	SW	CH1
BM-096 (M)	23	p.(Val282Leu)+ ClEx6/del	SW	CH2
BM-117 (F)	27	Del8bpEx3/del	SW	CH2
BM-122 (F)	36	c.293-13A/C>G/del	SW	CH1
BM-131 (F)	31	ClEx6/del	SW	CH1
BM-141 (M)	26	c.293-13A/C>G/del	SW	CH2
BM-182 (F)	32	p.(Gln319Ter)/del	SW	CH1
BM-037 (M)	23	p.(Ile173Asn)/del	SV	CH3
BM-051 (F)	22	p.(Pro31Leu)/del	SV	CH1
BM-100 (F)	23	p.(Pro31Leu)/del	SV	CH2
BM-111 (F)	23	p.(Ile173Asn)/del	SV	CH2
BM-154 (F)	33	p.(Ile173Asn)/del	SV	CH1
BM-005 (M)	18	p.(Val282Leu)/del	NC	CH2
BM-031 (M)	22	p.(Pro454Ser)/del	NC	CH1
BM-081 (M)	31	p.(Val282Leu)/del	NC	CH1

SV: Simple Virilizing, SW: Salt Wasting, NC: Non-Classic.

**Table 2 jcm-11-03818-t002:** Clinical findings in unrelated patients with monoallelic CH1 CAH-X and some parents carrying CH1 chimera.

Patient	Sex/Age	21OH Phenotype	Hypermobility Score ^a^	Skin Findings	Cardiac Findings	Other Clinical Features
BM-002	F/24	SW	4	Normal	Trivial mitral insufficiency	None
BM-014	F/41	SW	3	Skin laxity (neck and elbows), Easy bruising	Normal	Gastroesophageal reflux, hiatal hernia
BM-023	M/42	SW	2	Atrophic scarring	N/E	Chronic arthralgias, bilateral hallus valgus
BM-044	F/18	SW	8	Easy bruising	N/E	None
BM-067	M/19	SW	8	Thin skin with laxity, Easy bruising	N/E	None
BM-077	M/37	SW	3	Striae (abdomen and groin)	Normal	Chronic arthralgias, scoliosis
BM-122	F/36	SW	4	Easy bruising	Normal	Long uvula, gastroesophageal reflux, constipation, scoliosis
BM-131	F/31	SW	8	Normal	Normal	Gastroesophageal reflux, irritable bowel syndrome
BM-182	F/32	SW	7	Thin skin with laxity	Normal	None
BM-051	F/22	SV	7	Skin laxity	Normal	Subluxations, ples planus
BM-154	F/33	SV	6	Normal	Normal	Fibromialgias, hallus valgus, pes planus, long uvula
BM-031	M/22	NC	3	Normal	Normal	Chronic constipation
BM-081	M/31	NC	6	Normal	Atrial septum defect	Multiple dislocations, Long uvula
**Parent**						
BM-044 Mother	39	Not affected	4	Easy bruising	N/E	Chronic fatigue, anxiety, depression
BM-051 Father	47	Not affected	4	Normal	Normal	None
BM-067 Mother	44	Not affected	7	Normal	Normal	None

^a^ Hypermobility score was assessed by the Beighton scale [20]. SV: Simple Virilizing, SW: Salt Wasting, NC: Non-Classic.

**Table 3 jcm-11-03818-t003:** Clinical findings in unrelated patients with monoallelic CH2 CAH-X and some family members carrying CH2 chimera.

Patient	Sex/Age	21OHD Phenotype	Hypermobility Score ^a^	Skin Findings	Cardiac Findings	Other Clinical Features
BM-072	F/21	SW	8	Thin skin with laxity	Normal	None
BM-096	M/23	SW	7	Normal	Normal	Long uvula
BM-117	F/27	SW	6	Normal	Normal	Multiple dislocations, pes planus
BM-141	M/26	SW	3	Normal	N/E	Elongated uvula with midline crease, elbow dislocations, pes planus
BM-100	F/23	SV	3	Striae (groin)	Normal	Recurrent shoulder dislocations
BM-111	F/23	SV	4	Easy bruising	N/E	Arthralgias, elbow dislocations
BM-005	M/18	NC	4	Normal	Normal	None
**Parent**						
BM-005 Mother	F/48	Not affected	2	Normal	Normal	Hallus valgus, Gastroesophageal reflux
BM-072 Sister	F/17	Not affected	2	Normal	N/E	Normal

^a^ Hypermobility score was assessed by the Beighton scale [20]. SV: Simple Virilizing, SW: Salt Wasting, NC: Non-Classic.

**Table 4 jcm-11-03818-t004:** Clinical findings in the patient with monoallelic CH3 CAH-X and his family members carrying CH3 chimera.

Patient	Sex/Age	21OHD Phenotype	Hypermobility Score ^a^	Skin Findings	Cardiac Findings	Other Clinical Features
BM-037	M/23	SV	9	None	Mitral regurgitations	Ankle dislocation
**Parent**						
BM-037 Mother	F/47	Not affected	4	Striae (abdomen), Piezogenic papules	Normal	None
BM-037 Sister	F/21	Not affected	7	None	Normal	None
BM-037 Sister	F/16	Not affected	2	None	N/E	None

^a^ Hypermobility score was assessed by the Beighton scale [20]. SV: Simple Virilizing.

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
