# Peer review of "Prevalence of CAH-X Syndrome in Italian Patients with Congenital Adrenal Hyperplasia (CAH) Due to 21-Hydroxylase Deficiency"

_jcm, 2022, doi:10.3390/jcm11133818_

Round 1

Reviewer 1 Report

This is a comprehensive study on CAH-X focused in a population not previously described.  The group of patients under study has been  precisely defined and the proper methodology has been applied. The discussion is interesting, and the conclusions are relevant, at least, for those involved in the clinical follow-up of patients with congenital adrenal hyperplasia.

Two points should be improved/modified:

-      No mention has been done to the clinical examination of patients without CAH-X chimeras,

-      The second paragraph in the Discussion is confounding and would be better eliminated. Although a well-documented distinction of the concepts underlying the “30-kb deletion” by the different previous authors is presented, the authors try to justify the lowest frequency by Marino et al and Gao et al in this paragraph but, afterwards, they use the same argument to explain “a higher frequency” based on an underestimation of the denominator.

Author Response

We thank Reviewer 1 for comments and welcome your precious suggestions.

  1. In designing our study, we decided not to carry out a case-control study but to enroll only a selected population of patients carrying the CAH-X chimera. In this way, we focused our efforts on the clinical examination of a small group of patients. This allowed us to complete our study by optimizing our resources in light of the fact that the literature data available today widely report the clinical picture of patients with 21-hydroxylase deficiency carrying various genotypes.
  2. Although the second paragraph arises from a desire to explain the inconsistencies with the results obtained from previous studies, we understand that the discussion could be heavy and confusing. For this reason, we follow Reviewer 1's suggestion and decide to lighten the discussion by deleting the paragraph.

Reviewer 2 Report

Thank you for the opportunity to review the manuscript “Prevalence of CAH-X syndrome in Italian patients with Congenital Adrenal Hyperplasia (CAH) due to 21-hydroxylase deficiency” by Rosa Maria Paragliola et al. It is an interesting and relatively novel study on the two genetic disorders.   Please find my comments and suggestions:

1. Line 94: please add the name of the laboratory and institution.

2. I find the numbering of the patients in the Tables not clear. I would recommend putting them in some kind of order (e.g. SW, SV, NC) and number simply No 1, 2, 3.

3. In my opinion the text of the Results section is too detailed.

4. Again, the Discussion seems to include to much genetic details and therefore the essence of the text is not clear.

5. Conclusions- this section is too long, mostly should be part of Discussion.

Author Response

Reviewer 2

We thank Reviewer 2 for comments and welcome your precious suggestions.

  1. As suggested, the name of Laboratory and Institution was added.
  2. We welcome the suggestion and rework the Tables as directed. The patients were ordered according their phenotypes (SW, SV, NC).
  3. Our intent was to present the results more clearly. If it is not a problem, we prefer not to change the text of the results.
  4. We tried to lighten the discussion by eliminating some arguments concerning genetic data. However, the genetic details that we reported are an integral part of the essence of the paper and we cannot diminish their importance.
  5. As suggested, we reduced the Conclusions section and some of the text has been integrated into the Discussion.